# Sphingosine as a New Antifungal Agent against *Candida* and *Aspergillus* spp.

**DOI:** 10.3390/ijms232415510

**Published:** 2022-12-07

**Authors:** Fahimeh Hashemi Arani, Stephanie Kadow, Melanie Kramer, Simone Keitsch, Lisa Kirchhoff, Fabian Schumacher, Burkhard Kleuser, Peter-Michael Rath, Erich Gulbins, Alexander Carpinteiro

**Affiliations:** 1Institute of Molecular Biology, University Hospital Essen, University of Duisburg-Essen, 45122 Essen, Germany; 2Institute of Medical Microbiology, University Hospital Essen, University of Duisburg-Essen, 45122 Essen, Germany; 3Institute of Pharmacy, Freie Universität Berlin, 14195 Berlin, Germany; 4Department of Hematology and Stem Cell Transplantation, West German Cancer Center, University Hospital Essen, 45122 Essen, Germany

**Keywords:** sphingosine, sphingolipids, mitochondria, pulmonary aspergillosis, *aspergillus*, *candida*, yeast

## Abstract

This study investigated whether sphingosine is effective as prophylaxis against *Aspergillus* spp. and *Candida* spp. In vitro experiments showed that sphingosine is very efficacious against *A. fumigatus* and *Nakeomyces glabrataa* (formerly named *C. glabrata*). A mouse model of invasive aspergillosis showed that sphingosine exerts a prophylactic effect and that sphingosine-treated animals exhibit a strong survival advantage after infection. Furthermore, mechanistic studies showed that treatment with sphingosine leads to the early depolarization of the mitochondrial membrane potential (Δψm) and the generation of mitochondrial reactive oxygen species and to a release of cytochrome C within minutes, thereby presumably initiating apoptosis. Because of its very good tolerability and ease of application, inhaled sphingosine should be further developed as a possible prophylactic agent against pulmonary aspergillosis among severely immunocompromised patients.

## 1. Introduction

Sphingolipids are a group of lipids that share a sphingoid backbone and that modulate multiple intra- and intercellular functions, including apoptosis, proliferation, differentiation, and cell migration [1]. Sphingosine is an 18-carbon amino alcohol that can be generated from the cleavage of ceramide into sphingosine and a free fatty acid via ceramidases in lysosomes. Sphingosine may be reused in sphingolipid biosynthesis [2] or may be phosphorylated by sphingosine kinases 1 or 2 to sphingosine 1-phosphate (S1P), which may be degraded to hexadecenal and phosphoethanolamine via S1P lyase [1].

It has previously been shown that sphingosine has substantial in vitro and in vivo activity against both Gram-positive and Gram-negative bacteria, such as *Pseudomonas aeruginosa*, *Escherichia coli*, *Acinetobacter baumannii*, *Haemophilus influenzae*, *Burkholderia cepacia*, *Moraxella catarrhalis*, *Porphyromonas gingivalis*, *Staphylococcus aureus*, and even methicillin-resistant *S. aureus* (MRSA) [3,4,5,6,7,8,9].

Our in vivo studies showed that sphingosine is enriched in the epithelial cells of the trachea and large bronchi of healthy lungs and that it rapidly eliminates bacterial pathogens, whereas sphingosine concentrations decrease and ceramide accumulates in the respiratory tract of transgenic mice with cystic fibrosis. The inhalation of sphingosine prevented and eliminated pulmonary infection with *P. aeruginosa* in these mice [3,4,9].

*Aspergillus* spp. is commonly found in a variety of environments, especially in soil, water, air, plants, and grains. Only a few species are associated with major opportunistic infections in humans [10,11].

*A. fumigatus* is the most common airborne human pathogen worldwide, followed by *A. flavus*, *A. niger*, *A. terreus*, *A. nidulans*, and a number of species of the *Fumigati,* whose morphological features resemble those of *A. fumigatus* [12].

Typically, *A. fumigatus* conidia are eliminated by the immune system and do not cause serious disease in healthy persons. However, immunocompromised patients, including those with AIDS or severe neutropenia (<0.5/nL) and those who have undergone allogeneic hematopoietic stem cell or organ transplants and have been treated with corticosteroids, steroids, or other immunosuppressants, may experience colonization of the respiratory system, leading to pulmonary aspergillosis and eventual death [13]. In the immunosuppressed host, *A. fumigatus* conidia evade the mucociliary clearance of airways in the respiratory tract, and the germination of conidia in small bronchioles may cause life-threatening infection [14]. Despite treatment with approved drugs, such as polyene (e.g., amphotericin B) or azole antifungal agents, the mortality rate is still approximately 50% for patients who have undergone neutropenic or allogeneic hematopoietic stem cell transplant and approximately 90% for those with proven infection [15,16].

Guidelines suggest posaconazole as the primary antifungal prophylaxis for patients at high risk of invasive aspergillosis [17]. Posaconazole has been shown to be superior to fluconazole; however, probable or proven breakthrough invasive fungal infections still occur in approximately 10% of patients treated with posaconazole [18]. Echinocandins may be used alternatively in case of azole resistance [19]. However, the resistance of *Aspergillus* spp. to azoles and echinocandin can occur because of the use of these agents not only in clinical medicine but also in agriculture.

*Candida genus* yeasts are members of a fungal group that is a common part of the normal commensal microbial flora, colonizing the skin, oral cavity, vagina, and gastrointestinal tract of healthy persons. By altering the host microbiota or the host immune defense, *Candida* spp., such as *C. albicans*, *Nakeomyces glabrataa* (formerly named *C. glabrata*), *C. tropicalis*, *C. parapsilosis*, and *Pichia kudriavzevii* (formerly named *C. krusei*), can become pathogenic and cause severe superficial, mucosal, deep-seated, or systemic infections, especially among hospitalized patients [20,21].

Cutaneous candidiasis is the most common fungal infection among humans, typically involving almost any part of the skin and nails. Although rarely invasive, uncontrolled infections can become persistent and can impose considerable costs on healthcare systems [22].

In contrast, invasive candidiasis is a serious health problem that may cause intra-abdominal abscesses, peritonitis, osteomyelitis, and bloodstream infections (candidemia), especially among immunosuppressed patients. Overall, *C. albicans* is the most common fungal pathogen, but non-*albicans Candida* species such as *N. glabrataa*, *P. kudriavzevii,* and *C. parapsilosis* are collectively isolated in more than 50% of all candidemia cases worldwide [20,23].

Because the therapeutic options for, e.g., invasive aspergillosis and also some *candida* infections are limited, and because of the development of resistance to azoles, echinocandin, and amphotericin B [24,25], new therapeutic and prophylactic approaches are greatly needed. The present study was designed to determine whether sphingosine is suitable as a new antifungal treatment. We tested and characterized the efficacy of sphingosine in vitro under culture conditions and in a murine model of pulmonary aspergillosis.

## 2. Results

We determined the antifungal activity of sphingosine against twelve clinical and three reference isolates of *Aspergillus* spp. 48 h after infection (Table 1). All clinical and reference strains of *A. fumigatus* exhibited a minimum inhibitory concentration (MIC) of 2 μg/mL; the strain *A. niger* F19 exhibited a MIC of 4 μg/mL; and all isolates of *A. flavus*, *A. brasiliensis,* and *A. tubingensis* exhibited a MIC higher than 8 μg/mL. Moreover, we found that nearly all minimum fungicidal concentrations (MFCs) were equal to MICs except for two *A. fumigatus* clinical strains: *A. fumigatus* 2453, MFC 4 μg/mL, and *A. fumigatus* 2040, MFC 8 μg/mL (Table 1).

We further determined the antifungal activity of sphingosine against 13 clinical isolates of *Candida* spp. and against the reference strains of *C. albicans* ATCC90028, *N. glabrataa* DMS70614, *C. tropicalis* DSM1346, and *C. parapsilosis* ATCC22019 (Table 2). Of these, the planktonic cells of the *N. glabrataa* reference strain were the most susceptible to sphingosine (MIC, 1 μg/mL), followed by clinical strains of *N. glabrataa* (MIC, 2 μg/mL) and *P. kudriavzevii* (MIC, 4 μg/mL), whereas *C. albicans*, *C. tropicalis*, and *C. parapsilosis* planktonic cells were resistant, exhibiting a MIC higher than 8 μg/mL. In addition, the activity of sphingosine toward sensitive strains was also investigated in terms of MFC. The MFC values were equal to the MIC values for *N. glabrataa* DSM70614, *N. glabrataa* 196, *P. kudriavzevii* ATCC6258, and *P. kudriavzevii* 201. For the other strains, MFC values were higher than the corresponding MIC values (Table 2).

To illustrate the relationship between the concentration of sphingosine and its fungicidal effect over time, we exposed *N. glabrataa* DSM70614 strains to various concentrations of sphingosine (vehicle control, 0.5 × MIC, 1 × MIC, and 2 × MIC) for various time periods (0 to 24 h). To determine the number of viable cells, we determined the number of colony-forming units (CFU) per mL (Figure 1a). Sphingosine concentrations equal to 2 × MIC and 1 × MIC were found to be fungicidal (more than a 4-log reduction in the number of colonies) after 6 h or 10 h, respectively. In contrast, concentrations of 0.5 × MIC exhibited no inhibitory effect, and the resultant curve was approximately comparable to those for the solvent-treated vehicle control.

The quantification of viable cells by counting CFUs in an agar medium is not possible for multicellular filamentous fungi that may grow as only a single CFU when plated on agar. Therefore, XTT reduction assays were used as an alternative to a CFU-based method of determining the kinetics of killing *Aspergillus* strains.

To compare the cell viability values obtained by XTT assay with the number of colonies, we fitted a calibration curve (Figure 1b). A linear relationship was confirmed between the XTT value and an initial inoculum size between 10^5^ and 5 × 10^8^ conidia per mL (*R^2^* = 0.8973).

The effects of various concentrations of sphingosine on *A. fumigatus* ATCC46645 conidia over a 24 h period are shown in Figure 1c. Adding various concentrations of sphingosine (1×, 2× and 4 × MIC) to a culture medium containing 10^6^ conidia reduced formazan absorbance, which was roughly equivalent to a reduction in hyphal viability, according to the data generated from the calibration curve (Figure 1b). Within 10 h, approximately 90% of the conidia were killed by sphingosine at 1 × MIC (>1-log reduction), whereas 2 × MIC accomplished 99.9% killing (>3-log reduction) and 4 × MIC accomplished 99.99% killing (>4-log reduction), a finding demonstrating that sphingosine kills *A. fumigatus* in a time- and dose-dependent manner. As expected, no antifungal activity was found at a concentration of 0.5 × MIC.

To study the efficacy of sphingosine as an antifungal agent in an in vivo model, we used a murine model of invasive aspergillosis and administered sphingosine by inhalation. Groups of eight immunocompromised mice were infected with *A. fumigatus* via an intratracheal tube. The test group received sphingosine via inhalation at a dose of 500 µM in 1 mL volume of solvent for the sequential 14 days of treatment. As controls, immunosuppressed mice were inoculated with *A. fumigatus* and treated with an inhaled solvent plus a placebo. The survival rate of the sphingosine-treated mice was markedly better than that of the placebo-treated mice (Figure 2a). *A. fumigatus* ATCC48846 infection resulted in a mortality of 50% within 4 days and in weight loss (Figure 2b) in placebo-treated mice, whereas 100% of mice treated with sphingosine survived until the end of the observation period, a finding demonstrating that nebulized sphingosine prevents death caused by invasive aspergillosis in a murine model.

Two groups of immunosuppressed and infected mice (non-treated control and sphingosine-treated) were put to death immediately after infection (day 0) and four days after infection. The fungal burden of organisms isolated from the lungs was determined by counting CFUs after serial dilutions. We were able to detect and quantify *Aspergillus* conidia in both treated and non-treated groups with a CFU assay (Figure 2c). After 4 days of treatment with inhaled sphingosine, we found a statistically significant (*p* < 0.05) reduction in the fungal burden of nearly two log levels. In contrast, the placebo-treated mice did not exhibit a significant reduction in fungal burden (Figure 2c).

Galactomannan is a polysaccharide cell wall component of *Aspergillus* spp. that is released by fungal hyphae during invasive growth and can be detected by immunoassay as an additional method of quantifying fungal burden [27]. Therefore, to confirm our data, we quantified galactomannan in the bronchoalveolar lavage (BAL) fluid as an additional index of fungal burden. The galactomannan level, determined as the optical density index (ODI) in BAL fluid collected on day 4 after infection, was notably and statistically significantly lower (*p* < 0.05) for the group treated with inhaled sphingosine than for the placebo-treated control group (Figure 2g). Galactomannan-positive and -negative controls were available in the commercial kit.

When the lungs were removed on day 4 after infection, the lungs of the sphingosine-treated mice looked macroscopically similar to those of uninfected animals, whereas the lungs of the placebo-treated mice exhibited a distinctly denser inflammatory aspect (not shown). This finding was confirmed by the histochemical examination of tissue sections. In the lungs of the infected placebo-treated mice we detected large numbers of branching septate hyphal elements of *A. fumigatus,* along with prominent tissue necrosis and inflammation (Figure 2e), whereas in sphingosine-treated mice, this was not the case (Figure 2d).

Inflammation was quantified by threshold particle analysis with ImageJ software. The inflammation in the lungs after a dose of 500 μg sphingosine was similar to that seen in normal non-infected lungs from C57BL/6J mice. This level of inflammation was statistically significantly (*p* < 0.0001) less than the pronounced inflammation observed in the infected placebo-treated mice (Figure 2f). 

To determine whether treatment with inhaled sphingosine triggers apoptosis in mouse lung cells in vivo, we performed terminal transferase-mediated deoxyuridine triphosphate nick-end labeling (TUNEL) assays on lung sections. We observed no increased apoptotic signals in non-infected sphingosine-treated (Figure 3a) mice or in infected sphingosine-treated mice (Figure 3b). In contrast, the lungs of untreated *A. fumigatus*–infected mice showed a dramatic increase in the number of TUNEL-positive cells (Figure 3c) because of cell death caused by invasive aspergillosis. DNase I–treated tissue was used as a positive control (Figure 3d). These results clearly show that treatment with sphingosine did not mediate either apoptosis or toxicity in lung cells.

Because *Aspergillus* spp. grow in hyphae in liquid culture, single-cell analysis to investigate the mechanism of sphingosine toxicity is not possible. Therefore, we selected the reference strain *N. glabrataa* DSM70614 for further experiments aimed at elucidating the mode of action of sphingosine.

To determine the effect of sphingosine on cell wall integrity, we stained cells with calcofluor white and FUN1 (LIVE/DEAD Yeast Viability Kit, molecular probes). Calcoflour is a fluorescent dye that binds polysaccharides of the fungal cell wall, whereas FUN 1 is a two-colour fluorescent viability probe for yeasts. Metabolically active cells can compact the dye into orange-red cylindrical intravacuolar structures (CIVS), whereas dead cells or cells with intact membranes but with no metabolic activity exhibit bright green cytoplasmic fluorescence in the absence of fluorescent intravacuolar bodies.

Non-treated control cells were oval, and calcofluor dye was uniformly distributed on the cell wall. Orange-red FUN1 dye was compacted in CIVS, a finding indicating the intact metabolic activity of the controls, as expected (Figure 4a). *N. glabrataa* DSM70614 cells treated with sphingosine (2 × MIC) for 4 h did not form orange-red dye compacted in CIVS, a finding indicating that they were metabolically inactive (Figure 4b), although cell wall integrity remained intact.

Treating the cells with 70% *v*/*w* ethanol as the negative control led to a loss of calcofluor signal and to a lack of formation of orange-red dye compacted in CIVS, as expected (Figure 4c). These findings suggest that, although sphingosine can kill or suppress metabolic activity, it has no effect on cell wall integrity, a finding indicating that the killing mechanism of sphingosine does not directly affect the cell wall.

Next, we determined the effect of sphingosine on the plasma membrane. Propidium iodide (PI) is a nucleic acid–binding fluorescent probe that enters only cells with a compromised plasma membrane [28]. Therefore, we treated *N. glabrataa* DSM70614 cells for various time periods with increasing concentrations of sphingosine, stained them with PI, and counted PI-positive cells by flow cytometry (Figure 5a). We found that sphingosine exerts a time- and concentration-dependent effect on cell membrane integrity. Similar patterns of PI staining and death were recorded for amphotericin B–treated cells as positive controls (Figure 5a). However, sphingosine treatment at 0.5 × MIC did not increase PI fluorescence, a finding indicating that sub-MIC sphingosine does not affect cell membrane integrity.

Reactive oxygen species (ROS) play a crucial role as potent intrinsic stimuli for apoptosis in yeasts and other filamentous fungi. In the presence of ROS, the non-fluorescent H2DCFDA is rapidly converted to highly fluorescent 2′,7′-dichlorofluorescein (DCF) [29]. Therefore, we measured intracellular ROS by using the fluorescent indicator H2DCFDA in *N. glabrataa* DSM70614 exposed to various concentrations of sphingosine for various durations and then detected fluorescence by flow cytometry (Figure 5b). Our results show that the fluorescence intensity of the cells treated for 30 or 60 min with 1 × MIC sphingosine is dramatically higher than that of the untreated control cells and is even higher for the cells treated with 2 × MIC sphingosine.

Mitochondrial membrane potential (MtMP) is essential for maintaining the function of proton pumps and for producing cellular adenosine triphosphate (ATP) through oxidative phosphorylation; it plays a key role in mitochondrial function [30]. Loss of MtMP is a point of no return in the induction of apoptosis [31,32,33] and results in the release of cytochrome C and cell death [34].

To determine the effect of sphingosine on the MtMP of *N. glabrataa* DSM70614, we subjected the cells to 10, 60, or 120 min of treatment with sphingosine at various concentrations and then stained them with Rh123, which accumulates within energized mitochondria with intact MtMP. We found a statistically significant reduction in total fluorescence intensity (Figure 6a) after only 10 min of treatment with 1 × MIC sphingosine, a finding indicating that mitochondrial membrane depolarization is an early event in killing by sphingosine. Increasing the sphingosine dose to 2 × MIC or increasing the duration of treatment to 120 min does not significantly increase the effect. The cells treated with NaN_3_, which induces apoptosis by increasing the permeability potential of the mitochondrial membrane and inhibiting cytochrome oxidase in the mitochondrial electron transport chain [35,36], were used as positive controls.

Next, we evaluated the accumulation of mitochondrial ROS by using the MitoSOX Red reagent. Oxidation of MitoSOX Red by ROS, especially superoxide, produces red fluorescence, which can be detected by flow cytometric analysis and corresponds to the amount of ROS present in the mitochondria. We found that sphingosine treatment leads to an increase in mitochondrial superoxide levels, as indicated by a statistically significant shift in the fluorescence signal peak toward higher ROS levels than in untreated controls (Figure 6b). This effect was time- and dose-dependent.

The release of cytochrome C from mitochondria to the cytosol is a representative marker and hallmark of cellular apoptosis that may be initiated by ROS [37]. To determine the release of cytochrome C, we isolated fresh mitochondria from *N. glabrataa* and incubated them with various concentrations of sphingosine for 10, 30, or 60 min. The release of cytochrome C into the supernatant was detected by Western blotting (Figure 6c). Our findings show that treatment with sphingosine results in the release of cytochrome C to the cytosol in a dose- and time-dependent manner. The cells treated with 1 mM H_2_O_2_ were used as positive controls. Our results show that the action of sphingosine directly on mitochondria induces the release of cytochrome C in vitro, a finding supporting the notion that a direct effect on mitochondria may be the main mechanism of the fungicidal effect of sphingosine.

## 3. Discussion

Many studies have shown that sphingosine exerts antibacterial activity against Gram-positive and Gram-negative bacteria. Sphingosine has also been shown to prevent infections with some viruses [3,4,5,6,7,8,9,38]. Inhaled nebulized sphingosine eliminated *P. aeruginosa* and *S. aureus* in pulmonary-infected cystic fibrosis mice without producing severe toxic adverse effects [3,4,9].

In the present study, we found that sphingosine is effective against some of the strains tested (Table 1 and Table 2), especially *N. glabrataa* and *A. fumigatus* strains. The antifungal activity of sphingosine against *Aspergillus* spp. (Table 1) and *Candida* spp. (Table 2) was determined according to the European Committee for Antimicrobial Susceptibility Testing (EUCAST) protocol for yeasts and conidia-forming moulds. Interestingly, we found marked differences in the sensitivity to sphingosine between *Aspergillus* spp. whereas *A. fumigatus* is rather sensitive to sphingosine treatment, *A. niger*, *A. flavus, A. brasiliensis,* and *A. tubingenesis* are much less sensitive (Table 1). The background for the differences in sensitivity of the various strains of *Aspergillus* is outside the focus of this study and was not further investigated. Although *N. glabrataa* is rather sensitive to sphingosine treatment, *P. kudriavzevii*, *C. albicans*, *C. tropicalis*, and *C. parapsilosis* strains are less sensitive (Table 2). It is interesting to note that *N. glabrataa* is a weak producer of biofilm, *P. kudriavzevii* is an intermediate producer, and *C. albicans*, *C. tropicalis,* and *C. parapsilosis* are strong producers [39,40]; the ability to produce biofilm seems to be inversely correlated with sensitivity to sphingosine. However, our experiments were performed according to the EUCAST protocol and used planktonic cells; therefore, biofilm itself should not have been present at the time of sphingosine exposure and should not have caused the discrepancy. Nevertheless, we cannot exclude biofilm as a cause of the differential results because biofilm was not tested, and the presence of biofilm could not be completely excluded.

We further studied the efficacy of sphingosine as an antifungal agent in an in vivo model of invasive aspergillosis in immunosuppressed mice. Mice were infected with *A. fumigatus* ATCC46645. Sphingosine was applied daily by inhalation for 10 min; inhalation was initiated 3 days before infection. Therefore, our translational model closely mimics a prophylactic setting. Prophylactic treatment with inhaled sphingosine starting 3 days before *Aspergillus* infection and continuing for as long as 14 days resulted in 100% survival and in the resolution of the infection, whereas the lethality among the mice treated with the inhaled placebo reached the dropout burden of 50%. We further quantified the *Aspergillus* burden 1 h after infection and 4 days after infection in sphingosine-treated and placebo-treated mice. Although one hour after infection, the number of detectable conidia in sphingosine-treated and placebo-treated animals was approximately the same, as early 4 days after infection, there was already a clear difference: the CFU assay showed a 99% reduction in detectable *Aspergillus*, a finding indicating a marked effect of sphingosine. We confirmed these findings by measuring the galactomannan concentrations in BAL fluid from infected mice as a surrogate parameter for fungal burden. In additional experiments (data not shown), we initiated sphingosine treatment 3 days after infection. We could not detect any protective effect improving survival among mice infected by the application of conidia. It is, therefore, possible that the effect of sphingosine is essentially limited to the killing of conidia. The therapeutic effect of sphingosine after later application (from day 3 of infection) conferred no measurable survival advantage compared to placebo (not shown), perhaps because the mice in the selected setting may have already been too sick and the disease too far advanced, having passed a point of no return. On the other hand, we can speculate that the therapy may have been ineffective because of the formation of hyphae with their invasive growth; the inhaled sphingosine may not have reached the area of severe infection because of the associated disturbance in aeration.

However, it must be assumed that our murine model of invasive aspergillosis may have several limitations in comparison to the human setting. First, the spontaneous course of the disease is very short, achieving the dropout burden of 50% lethality among placebo-treated mice after only 4 days. Among humans, invasive aspergillosis, even among immunosuppressed patients, develops over several weeks to months. Second, because of the direct intratracheal application or nebulization, the lung is affected very rapidly. In contrast, in the human situation, invasive aspergillosis generally does not affect the entire lung [41]. Future studies are necessary for establishing a curative model.

Our treatment protocol was very well tolerated; the mice exhibited no visible adverse effects that would have been attributable to inhalation treatment with sphingosine. TUNEL assays performed on lung tissue sections after 7 days of sphingosine inhalation detected no cell death. These findings are in line with previous work in minipig and murine models of sphingosine inhalation in other contexts [42,43,44]. In conclusion, we found that sphingosine is effective in a murine model of pulmonary aspergillosis, at least in a prophylactic setting, and that the therapy is well tolerated without detectable adverse effects.

Yeasts such as *C. albicans*, *N. glabrataa*, *P. kudriavzevii*, *C. tropicalis,* and *C. parapsilosis* are the most common cause of vaginal or mucosal infections and may, among immunosuppressed patients, enter the bloodstream and cause deep tissue or systemic infections [20,21]. Systemically applied azoles, echinocandin, and locally applied polyenes are used as prophylaxis against various diseases. Our in vitro findings show that *N. glabrataa* and *P. kudriavzevii*, both known to have intrinsic resistance to azoles [40], are sensitive to sphingosine treatment. It is quite conceivable that prophylactic or therapeutic treatment by local application of sphingosine in creams, for example, could have additional treatment applications. However, the present project did not pursue this option, and the possibility remains untested. 

The commonly used antifungal drugs can be classified on the basis of their site of action. The fungistatic azoles inhibit ergosterol biosynthesis via the cytochrome P450 enzyme 14-α demethylase, which is involved in the conversion of lanosterol to ergosterol [45]. The depletion of membrane ergosterol by the use of azoles causes cell membrane disruption and growth inhibition [45,46]. Polyenes mainly include amphotericin B; natamycin and nystatin exhibit fungicidal activity primarily by binding to ergosterol and forming pores in the plasma membrane that promote leakage of intracellular ions and in cell death [47,48,49]. The inhibition of DNA and RNA synthesis is related to the action of pyrimidine analogues such as 5-flucytosine (5-FC), which is converted to 5-fluorouracil and then to 5-fluorouridylic acid by uridine monophosphate (UMP) pyrophosphorylase [47,48]. The fungal cell wall containing chitin and β-glucan is another target of many antifungal drugs. It is known that treatment with members of the echinocandin family, including caspofungin, micafungin, and anidulafungin, can inhibit cell wall synthesis by acting as non-competitive inhibitors of -1,3 glucan synthase, which is required for glucan synthesis [50,51].

Regarding our studies aimed at elucidating the mode of action of sphingosine, we selected the reference strain of *N. glabrataa* DSM70614, which is sensitive to sphingosine treatment (Table 2) as an experimental system to elucidate the mode of action of sphingosine. We found that treatment with sphingosine does not affect cell wall integrity. We found that the cell membrane is disrupted by staining with PI, and this disruption increased substantially during incubation within 6 h. In addition, we found a marked increase in the amount of cellular ROS within one hour after treatment. We also found that treatment with sphingosine leads to the early depolarization of the MtMP (Δψm) and to mitochondrial ROS generation within minutes. We also found a direct effect of sphingosine on mitochondria leading to the release of cytochrome C as a hallmark of apoptosis. In summary, our findings suggest that the mode of action of sphingosine on fungi is due to a direct effect on mitochondria and that the disruption of the cell membrane may be a later secondary event in the frame of apoptosis. 

Cardiolipin consists of two phosphatidyl moieties appended to a glycerol backbone. The lipid differs from all other phosphoglycerides in that it carries four fatty acids instead of two. Cardiolipin is found in the inner membrane of mitochondria and in bacterial membranes but not in eukaryotic cell membranes. It has been shown that the presence of the protonated NH_2_ group in sphingosine, which is an amino alcohol, is required for sphingosine’s bactericidal activity [52]. The protonated NH_2_ group of sphingosine binds to the highly negatively charged lipid cardiolipin in bacterial plasma membranes. *E. coli* or *P. aeruginosa* strains that lack cardiolipin synthase are resistant to sphingosine, a finding finally indicating that the binding of sphingosine and cardiolipin is required for sphingosine toxicity [52]. Presumably, this clustering results in the rapid permeabilization of the plasma membrane and in bacterial cell death. As mentioned above, cardiolipins are also present in the inner mitochondrial membrane so that the data on the mechanism of action generated in bacteria may be transferable to mitochondria. This hypothesis is supported by the fact that effects indicating the permeabilization of the inner mitochondrial membrane become clearly visible within a few minutes of sphingosine treatment. The present project did not investigate the molecular effect of sphingosine on mitochondria, but such effects should be investigated in future projects.

If the toxicity of sphingosine is due to its effect on cardiolipins of the inner mitochondrial membrane, the question naturally arises as to how the specificity of the effect could be achieved in the murine model. No toxicity was detected in normal lung tissue. In line with our results, previous studies have shown no relevant toxicity in the bronchial epithelium after inhalation of up to 1000 µM sphingosine [42]. The exact reason for this finding is unknown, but it could be that inhaled sphingosine micelles are trapped in the mucus and can kill pathogens but not penetrate the epithelium itself. In accordance, it has been shown that a large portion of administered sphingosine remains in the mucus on top of the epithelial cell layer and is not taken up [44]. It is also conceivable that bronchial epithelium metabolizes sphingosine more rapidly than other tissues because of the increased expression of, e.g., sphingosine kinases, which reduce the toxicity of sphingosine. However, this hypothesis has not been investigated so far and remains speculative. The reason for sphingosine resistance in normal bronchial cells requires further definition.

Our data suggest that the therapeutic targets of sphingosine are particularly the mitochondria of fungi. This distinguishes sphingosine from the other drugs used in routine clinical practice, such as azoles, polyenes, and echinocandins. Thus, it can be speculated that prophylactic treatment with inhaled sphingosine will not lead to the development of resistance against the drugs already used in clinical routine. This could therefore be an important advantage for the use of sphingosine as a prophylactic agent, since in the case of a breakthrough infection, the whole range of proven substances would still be available. However, this was not investigated in the present study and thus remains speculative.

In summary, the results of the present study show that sphingosine is effective in killing *A. fumigatus*. Using a murine model of pulmonary aspergillosis, we found that prophylactically applied inhaled sphingosine protects against the lethal course of pulmonary aspergillosis. Our findings suggest that the mechanism of action is a direct effect on mitochondria, but additional experiments are needed to confirm this hypothesis. Because of its very good tolerability and ease of application, sphingosine should be further investigated as a possible prophylactic agent against pulmonary aspergillosis among severely immunocompromised patients.

## 4. Materials and Methods

### 4.1. Fungal Strain and Culture Conditions

*N. glabrataa* DSM70614, *P. kudriavzevii* ATCC6258, *A. fumigatus* 46645, and other standard and clinical samples were generously donated by the Institute of Medical Microbiology, University Hospital Essen, University of Duisburg-Essen. All included strains are listed in Table 1 and Table 2.

Yeast isolates were identified by matrix-assisted laser desorption/ionization time-of-flight (MALDI-TOF) mass spectrometry (VITEK; bioMérieux, Nürtlingen, Germany). *A. fumigatus* was identified by the typical macro-and micromorphology and growth at 50 °C. Bacteria other than *A. fumigatus* spp. were identified on the basis of their morphology and on analysis of the internal transcribed spacer (ITS) 1 and 2 DNA sequences. 

All of the strains were stored as a 50% glycerol stock at −80 °C and were grown in liquid yeast extract peptone dextrose (YPD) (Gibco, Grand Island, NY, USA) media. For planktonic susceptibility testing, cells were grown on Sabouraud dextrose agar (SDA) plates (Oxoid, Wesel, Germany) at 30 °C for 24 h; 2 to 3 colonies were used for the cell inoculum. Medium containing Roswell Park Memorial Institute (RPMI) 1640 (Sigma-Aldrich, St. Louis, MO, USA) + 3-(N-morpholino) propanesulfonic acid (MOPS) (Sigma-Aldrich) + 2% glucose (Sigma-Aldrich) (pH 7.0) was used for all susceptibility tests. 

### 4.2. Planktonic Susceptibility Testing

To determine MICs, we used the broth microdilution method for yeast and conidia-forming moulds, according to the EUCAST guidelines [53]. The MIC of sphingosine is the lowest concentration that causes growth inhibition of 90% or higher compared to the drug-free control.

The antifungal agent was diluted in 10% octylglucopyranoside (OGP; Sigma-Aldrich) and RPMI 1640 for the preparation of stock solutions. The test ranges for broth microdilution of sphingosine (d18:1) (Avanti Polar Lipids, Alabaster, AL, USA) were 0.06 to 32 mg/L.

Two-fold dilutions of sphingosine were prepared in RPMI 1640 + MOPS + 2% glucose (pH 7.0), and 100 µL was mixed with 100 µL of cell suspensions in water (final working inoculum, 2–5 × 10^5^ cells per mL) in 96-well plates (Sarstedt, Nümbrecht, Germany). The plates were incubated without agitation at 37 °C in ambient air for 24 or 48 h, and the growth of *Aspergillus* strains was evaluated visually, whereas for *Yeast* samples, the absorbance at 530 nm was measured (Table 1 and Table 2).

To obtain the MFC, we spread 200-μL sphingosine cell suspensions on SDA plates (Oxoid, Wesel, Germany), as described above. The plates were incubated at 37 °C for 48 h. The MFC was defined as the lowest drug concentration that yielded three or fewer colonies (i.e., 99.0% to 99.5% killing activity).

### 4.3. Planktonic Growth-Inhibition Kinetic Studies for the N. glabrataa (DSM70614) Strain

Before initiating the growth-inhibition kinetic studies, we determined the antifungal carryover effect by using the method described by Klepser et al. [54]. Additionally, all of the samples were plated as a single spot on SDA plates (Oxoid, Wesel, Germany) and were allowed to diffuse into the agar for 10 min before spreading. After the plate had dried, streaking was performed as described by Lignell et al. [55]. No carryover antifungal effect was observed.

Before the tests were performed, *N. glabrataa* (DSM70614) was grown for 24 h at 35 °C on PDA plates (Oxoid, Wesel, Germany). The inoculum was quantified with a haemocytometer (Neubauer Improved counting chamber; Paul Marienfeld GmbH, Lauda-Königshofen, Germany) and was adjusted to 1 × 10^6^ to 5 × 10^6^ CFU/mL. To 1 mL of the adjusted fungal suspension, we added 9 mL of either RPMI 1640 medium buffered with MOPS buffer and 2% glucose (control) or a solution of growth medium plus an appropriate amount of sphingosine. The cells were incubated at 35 °C and 200 rpm for various time periods. We then serially diluted 100-µL aliquots and streaked them on potato dextrose agar (PDA) plates for the determination of CFUs (Figure 1a). All of the experiments were conducted in triplicate.

### 4.4. Quantification of Viable Aspergillus conidia Using XTT [26]

Various conidial concentrations were prepared in RPMI 1640 + MOPS + 2% glucose (pH 7.0), as indicated (Figure 1b). We added 200 µL of each to flat-bottomed 96-well plates. We added 50 µL of XTT (Santa Cruz Biotechnology, Dallas, TX, USA) (1 mg) and menadione (Sigma-Aldrich) (0.17 mg) solution and incubated the plates at 37 °C for 2 h. The aliquots were serially diluted and streaked on a PDA plate for CFU determination.

The optical density of each inoculum concentration was measured at 450 nm. A standard calibration curve was plotted with log CFU per mL and the respective absorbance within the XTT reduction assay (Figure 1b), and this curve was used as a reference for the quantification of viable cells.

### 4.5. Growth-Inhibition Kinetic Studies with XTT Reduction Assay for A. fumigatus (ATCC46645)

The XTT reduction assays were performed for time–kill kinetics, as previously described [56], with slight modifications. Briefly, separate microplates were used for each incubation timepoint. We added 100 μL of double-concentrated sphingosine (or negative control) to *A. fumigatus* conidia in double-concentrated RPMI 2% glucose medium as indicated (Figure 1c).

All of the plates were incubated at 37 °C without agitation for up to 24 h. At the indicated timepoints 2 h before the indicated incubation time, 50 µL of XTT–menadione solution was added to each well. After 2 h of incubation, formazan absorbance was measured at 450 nm with a microplate reader (BMG Labtech, Offenburg, Germany). All experiments were performed in triplicate.

### 4.6. Cell Wall Staining

The effect of sphingosine on the cell wall of *Candida* was determined by calcofluor white staining, as previously described [57,58,59], with a few modifications. *N. glabrataa* DSM70614 was cultivated in YPD broth at 35 °C up to the mid-exponential phase. We treated 2 to 5 × 10^5^ cells per mL for 4 h (37 °C, 200 rpm) with two separate concentrations of sphingosine at a final concentration equal to 1 × MIC or 2 × MIC. After incubation, the cells were harvested, washed with 2% D-(+)-glucose (Sigma-Aldrich) containing 10 mM Na-HEPES (Sigma-Aldrich), pH 7.2, and adjusted to a concentration of 5 × 10^7^ cells/mL. *Candida* incubated in 70% ethanol for 4 h was used as a positive control [60].

The samples were stained for viability with LIVE/DEAD Yeast Viability Kit (ThermoFisher Scientific, Waltham, MA, USA) containing two fluorescent probes (FUN 1) and cell wall staining (calcofluor white M2R), according to the manufacturer’s instructions. Concisely, 100 µL of the yeast suspension containing 10 μM FUN1 was mixed with 100 μL calcofluor white M2R (final concentration, 25 μM). The cells were incubated for 30 min at 30 °C in the dark and were analyzed by fluorescence microscopy.

### 4.7. Plasma Membrane Disruption Assay Using Propidium Iodide

*N. glabrataa* DSM70614 was cultivated and treated as mentioned above with various concentrations of sphingosine as indicated or with 2 μg/mL amphotericin B as positive controls. After incubation for various time periods, the cells were washed 3 times in phosphate-buffered saline (PBS) and incubated with PI (1 µg/mL final concentration) for 5 min at room temperature. The MFI values of 10,000 events were measured with an Attune NxT flow cytometer (ThermoFisher Scientific).

### 4.8. Intracellular ROS Formation Assay 

The endogenous levels of ROS were assessed with 2′,7′dichlorodihydrofluorescein diacetate (H_2_DCFDA; ThermoFisher Scientific) [61,62]. Briefly, *N. glabrataa* DSM70614 cells were cultivated and exposed to various concentrations of sphingosine at 37 °C, along with 50 µM tert-butyl-hydrogen peroxide (TBHP) and 10 µg/mL antimycin A as positive controls. At various timepoints, the aliquots were removed and stained with 10 μM DCFH-DA at 37 °C for 30 min. After three washes with PBS, the fluorescence intensities (excitation/emission at 485/535 nm) of the cells were measured with an Attune NxT flow cytometer (ThermoFisher Scientific).

### 4.9. Determination of Mitochondrial Membrane Potential (ΔΨm)

The disruption of MtMP was assessed by staining with Rhodamine (Rh)-123, a fluorescent probe that is induced via mitochondrial energization [63,64,65]. *N. glabrataa* DSM70614 cells were cultivated and exposed to various concentrations of sphingosine at 37 °C. At various timepoints, the cells were harvested and stained with Rh-123 (25 µM for 10 min) and then washed three times with PBS. The mean intensity values of 10,000 events were measured per sample with an Attune NxT flow cytometer (ThermoFisher Scientific). Treatment with 20 mM sodium azide (NaN_3_) was used as a positive control. After staining with Rh-123, cells were washed three times with PBS, and the MFI values of 10,000 events were measured per sample with an Attune NxT flow cytometer (ThermoFisher Scientific).

### 4.10. Measurement of Mitochondrial ROS Levels

To detect mitochondrial ROS in live cells, we used MitoSOX Red dye (ThermoFisher Scientific) according to the manufacturer’s instructions [61,65,66]. *N. glabrataa* DSM70614 cells were cultivated and exposed to various concentrations of sphingosine at 37 °C. They were harvested at various timepoints by centrifugation at 10,000 rpm for 5 min and were incubated for 15 min at 37 °C in 1 × Hank’s Balanced Salt buffer without phenol red containing 5 µM MitoSOX Red, protected from light. The cells were washed three times with prewarmed buffer, and the fluorescent cells were counted with a BD FACSCalibur flow cytometer (BD, Franklin Lakes, NJ, USA).

### 4.11. Cytochrome C Release from N. glabrataa isolated Mitochondria

Isolated and functional mitochondria were used to investigate cytochrome C release as described, with minor adjustments [67,68,69]. In brief, *N. glabrataa* DSM70614 cells were cultured in YPD medium for 16 h at 30 °C, collected by centrifugation at 3000× *g*, and washed twice with distilled water. They were then resuspended in 2 mL DTT buffer per gram wet weight (100 mM Tris/H_2_SO_4_ [pH 9.4], 10 mM dithiothreitol) and incubated for 20 min at 30 °C. The cells were washed twice with 20 mM potassium phosphate (pH 7.4), 1.2 M sorbitol buffer, after which 5 mg zymolyase-20T per gram (wet weight) was added. The cells were then incubated for 30 min at 30 °C with slight agitation to form spheroplasts. *Candida spheroplasts* were washed with 10 mM Tris/HCl (pH 7.4), 0.6 M sorbitol, 1 mM ethylenediaminetetraacetic acid (EDTA), and 0.2% (*w*/*v*) bovine serum albumin (BSA) buffer; they were then homogenized in a prechilled glass homogenizer. Unbroken cells, nuclei, and large debris were pelleted by centrifugation for 5 min at 1500× *g* at 4 °C, and mitochondria were purified with sucrose density gradient centrifugation (60%, 32%, 23%, and 15% [*w*/*v*] sucrose in 1 mM MOPS/potassium hydroxide [KOH; pH 7.2] and 1 mM EDTA) at 134,000× *g* for 1 h.

Isolated mitochondria (0.9 mg of protein per mL) were incubated for 10, 30, or 60 min in the presence or absence of sphingosine with 1 mM H_2_O_2_ (positive control). After incubation, mitochondria were pelleted by centrifugation at 12,000× *g* at 4 °C for 15 min. The supernatant and pellet fractions were subjected to sodium dodecyl sulfate–polyacrylamide gel electrophoresis (SDS-PAGE) and analyzed by Western blotting with anti-cytochrome C antibody (BD Bioscience, Catalog # 556433). Horseradish peroxidase-linked goat anti-rabbit immunoglobulin G (Cell Signalling Technology, Danvers, MA, USA; Catalog # 7074) was used as the secondary antibody. The intensity of signals from Western blot analysis was quantified with a Typhoon FLA 9000 biomolecular imager (Cytiva, Marlborough, MA, USA).

### 4.12. Animal Model

A low-dose inhalational model of murine pulmonary aspergillosis was used to test the efficacy of sphingosine in vivo. Immunosuppressed mice were infected intratracheally, as previously described [70]. The breeding and experimental procedures were approved by the State Agency for Nature, Environment and Consumer Protection (LANUV), approval number AZ 81-02.04.2019.A153, NRW in Düsseldorf, Germany.

Briefly, C57BL/6J mice weighing 18 to 20 g each were immunosuppressed with 250 mg/kg of intraperitoneal cyclophosphamide (Baxter Oncology, Halle, Germany) and 200 mg/kg subcutaneous cortisone acetate (Sigma-Aldrich) on day −2 relative to infection (day 0). They were given a second dose of cyclophosphamide (200 mg/kg) and cortisone acetate (200 mg/kg) at day +3.

From day −3 to the end of the study, the mice were given 50 ppm of the antibiotic Baytril (Bayer, Leverkusen, Germany) via their drinking water to prevent bacterial infections.

### 4.13. Preparation and Administration of A. fumigatus

*A. fumigatus* (strain ATCC46645) conidia were collected in 0.1% Tween (Sigma-Aldrich) in PBS from 4- to 5-day-old cultures on PDA plates, filtered through a Filcons syringe fittings filter (10-μm pore size; BD Biosciences), and counted under a haemocytometer. For infection, the mice were anesthetized with 87.5 µg/g ketamine (MEDISTAR, Ascheberg, Germany) and 12.5 µg/g xylazine (Serumwerk Bernburg AG, Bernburg, Germany). After intubation, fungal forms were administered intratracheally at doses of 2 to 5 × 10^4^ conidia in 50 μL saline per mouse. The control mice received normal saline only.

For treatment studies, the mice were immobilized in a restrainer, and the clinically approved nebulizer (PARI BOY SX inhalation device; PARI, Starnberg, Germany) was used to deliver sphingosine/0.5% *w/v* OGP or 0.5% *w/v* OGP alone as a control. The daily inhalation treatment started on day −3 with the administration of 1 mL suspension over 10 min for the previously determined times. At day 0, inhalation was performed 1 h before infection with *A. fumigatus*.

The *A. fumigatus* burden of the mice was determined with a scoring system that included body weight, pulmonary symptoms, general condition changes, and behavioral changes. The burden was manifested by weight reduction and behavioral changes, so a weight loss of 10% to 19% or more and self-isolation with lethargy were considered the endpoints of the analysis, and the respective animals were immediately put to death with CO_2_ fumigation.

### 4.14. Fungal Burden in Mouse Lungs

The mice were pre-treated with cyclophosphamide and cortisone acetate, infected with fungal forms, and treated or not treated with inhaled sphingosine, as described above. The mice were put to death by CO_2_ fumigation immediately after infection (day 0) or on day 4 after infection. At each timepoint, two immunosuppressed but uninfected mice were included as negative controls [71,72]. The lungs were removed by sterile dissection, added to 2 mL of NaCl 0.85% *w*/*v*, and homogenized in a gentleMACS™ Tissue Dissociator (Miltenyi Biotech, Bergisch-Gladbach, Germany). After filtration through a 70-µm Falcon cell strainer (Fisher Scientific, Schwerte, Germany), the supernatants were serially diluted, spread on PDA plates, and incubated for 24 h at 37 °C. The *A. fumigatus* colonies were counted in triplicate.

### 4.15. Galactomannan Enzyme Immunoassay

The mice were pre-treated with cyclophosphamide and cortisone acetate, infected with fungal forms, and treated or not treated with inhaled sphingosine, as described above. The mice were put to death by CO_2_ fumigation immediately after infection (day 0) or on day 4 after infection. At each timepoint, two immunosuppressed but uninfected mice were included as negative controls. BAL was performed via a tracheal catheter, and the BAL fluid was collected by washing the airways twice with 0.5 mL of ice-cold sterile PBS [73]. Galactomannan was quantified in 300-µL aliquots of BAL fluid with Platelia Galactomannan EIA kits (Bio-Rad, Redmond, WA, USA) according to the manufacturer’s instructions. Results were expressed as optical density index (ODI).

### 4.16. Cytokine Detection

Enzyme-linked immunosorbent assays (ELISAs) were used to measure cytokine production in lung homogenate supernatants. Tumor necrosis factor (TNF)-α and interferon (IFN)-γ were assayed with commercially available antibody pairs and standards (BioLegend, San Diego, CA, USA) according to the manufacturer’s instructions.

### 4.17. In Situ Cell Staining

For histology, the left lungs were removed and immediately fixed in 4% paraformaldehyde (PFA) for 24 h. The samples were then dehydrated, embedded in paraffin, sectioned into 3- to 4-µm slices, and stained with periodic acid–Schiff reagent (PAS) (Carl Roth GmbH, Karlsruhe, Germany). In situ TUNEL assays were performed with an in situ Cell Death Detection kit, TMR red (Roche Diagnostics, Mannheim, Germany). All of the steps were performed according to the supplier’s instructions. Briefly, the paraffin-embedded sections were dewaxed, rehydrated, permeabilized in 0.1 M Na-citrate buffer pH 6.0 (microwave irradiation, 5 min, 350 W), and washed twice with PBS. The sections were then incubated for 60 min at 37 °C in a humidified chamber with a labeling solution containing terminal deoxynucleotidyl transferase. They were then washed twice with PBS and embedded with Mowiol (Carl Roth GmbH). The images were analyzed with Leica Confocal Software version 2.61 (Leica Microsystems, Mannheim, Germany). Tissue sections treated with DNAse I (1 mg/mL in Ca^2+^/Mg^2+^ buffer; Roche Diagnostics) were used as positive controls.

### 4.18. Quantitation of Lung Inflammation

Formalin-fixed sections of lung tissue were cut and stained with PAS. ImageJ (http://rsbweb.nih.gov/ij/, accessed on 1 August 2022) was used to calculate the area and pixel value statistics of user-defined selections and thereby to quantitate lung inflammation in tissue sections (*n* = 3) [72].

### 4.19. Statistical Analysis

The data are expressed as mean ± SD and were analyzed with GraphPad Prism 8 software (GraphPad Software, San Diego, CA, USA). The statistical details of the experiments can be found in the figure legends.

## Figures and Tables

**Figure 1 ijms-23-15510-f001:**
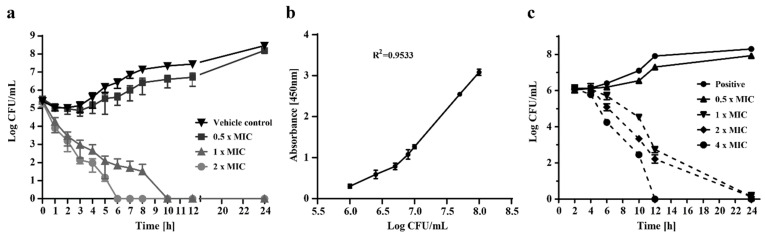
Sphingosine inhibits the growth of and kills *N. glabrataa* DSM70614 and *A. fumigatus* (ATCC46645) in a time- and dose-dependent manner. (**a**) We treated 1 to 5 × 10^5^ colony-forming units (CFU) per mL *N. glabrataa* DSM70614 with 3 separate concentrations of sphingosine (0.5×, 1×, or 2× minimum inhibitory concentration [MIC]) and a vehicle control. The log CFU per mL for all groups was determined at time 0 and at subsequent time points up to 24 h after as indicated. (**b**) Serial dilutions of *A. fumigatus* conidia were prepared, and XTT colorimetric intensity at 450 nm was measured after 2 h incubation. XTT activity was linearly associated with CFU counts of more than 10^5^ colonies per mL. A formula was calculated for a standard curve and the correlation coefficient [26]. (**c**) We treated 1 to 2.5 × 10^6^
*A. fumigatus* (ATCC46645) conidia per mL with various concentrations of sphingosine and then measured their viability with an XTT menadione solution, as indicated. CFUs were counted according to the standard curve. Shown are means ± standard deviation, *n* = 3.

**Figure 2 ijms-23-15510-f002:**
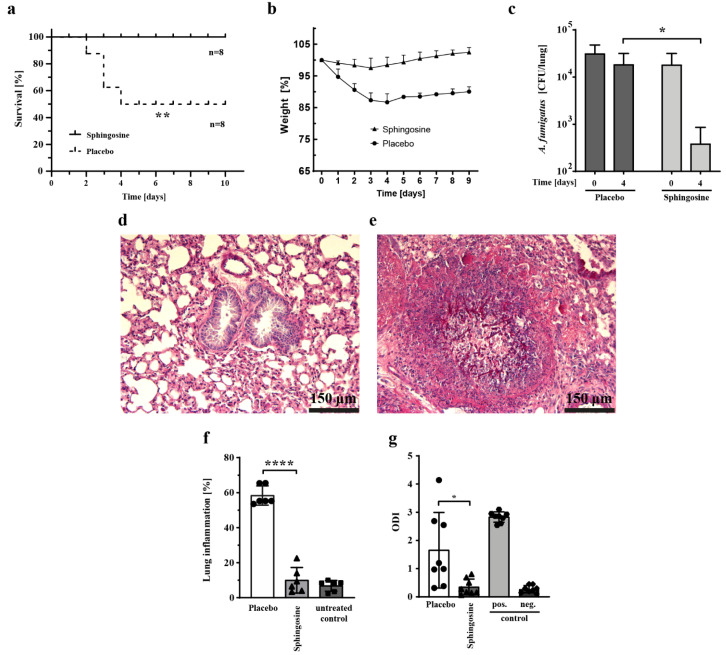
Prophylactic inhalation of sphingosine leads to a statistically significant reduction in fungal burden and prevents death in a murine model of pulmonary aspergillosis. C57BL/6J mice were immunosuppressed with 250 mg/kg of intraperitoneal cyclophosphamide and 200 mg/kg subcutaneous cortisone acetate on day 2 and on day +3. From day −3 to the end of the study, mice were given the antibiotic Baytril. For infection (day 0), mice were anesthetized and intubated, *A. fumigatus* (ATCC46645) forms were administered intratracheally (5 × 10^4^ in 50 μL 0.9% NaCl). For sphingosine inhalation, mice were immobilized in a restrainer and were given 1 mL of 500 µM sphingosine/0.5% octylglucopyranoside (OGP) by inhalation daily via a PARI BOY SX inhalation device, starting on day −3. (**a**) Kaplan–Meier survival curve, *n* = 8 per group. ** *p* < 0.01 (log-Rank test). (**b**) Mice were weighed daily, and the weight of each mouse from day 0 was set equal to 100%. Plotted are the respective relative weights of the living mice over time. (**c**) Immediately after infection (2 × 10^4^ conidia in 50 μL 0.9% NaCl) (day 0) and day 4 after infection, mice were put to death by CO_2_ fumigation, and lungs were removed by sterile dissection and homogenized. *A. fumigatus* CFUs were counted after serial dilutions and were spread on potato dextrose agar (PDA) plates. (Counts were performed in technical triplicates of *n* = 8 mice each. Shown are means ± standard deviation. * *p* < 0.05, Student’s *t*-test). (**d**,**e**) Lungs from mice treated as indicated were removed on day 4, fixed in 4% paraformaldehyde (PFA), dehydrated, embedded in paraffin, sectioned into 3- to 4-µm slices, and stained with periodic acid–Schiff reagent. Shown is one representative image from *n* = 6 mice. (**f**) Lungs from mice treated as indicated were removed on day 4, fixed in 4% PFA, dehydrated, embedded in paraffin, sectioned into 3- to 4-µm slices, and stained with periodic acid–Schiff reagent. ImageJ (http://rsbweb.nih.gov/ij/, accessed on 1 August 2022) was used to calculate the area and the pixel value statistics of user-defined selections and thereby to quantitate lung inflammation in tissue sections (*n* = 6 for treated animals, *n* = 3 for untreated controls). Shown are means ± standard deviation. **** *p* < 0.0001, Student’s *t*-test. (**g**) Bronchoalveolar lavage fluid was collected after 4 days of inhalation of sphingosine or solvent. Galactomannan antigen was detected in bronchoalveolar lavage fluid with Platelia Galactomannan EIA kits, and levels were expressed as optical density index (ODI) (*n* = 8). Shown are means ± standard deviation. * *p* < 0.05, Student’s *t*-test).

**Figure 3 ijms-23-15510-f003:**
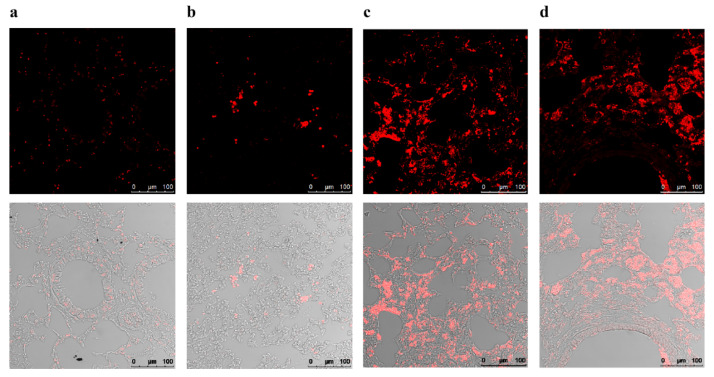
Inhalation of sphingosine does not affect normal lung tissue as measured by terminal deoxynucleotidyltransferase-mediated deoxyuridine triphosphate nick-end labeling (TUNEL) assay. Differentially treated mice were given 1 mL of 500 µM sphingosine/0.5% octylglucopyranoside (OGP) by inhalation daily over 7 days as indicated. Mice were put to death by CO_2_ fumigation on day 8. Lungs were fixed in 4% PFA, dehydrated, embedded in paraffin, sectioned into 3- to 4-µm slices, and further processed for TUNEL assay. (**a**) No pre-treatment. (**b**) Mice were immunosuppressed with 250 mg/kg intraperitoneal cyclophosphamide and 200 mg/kg subcutaneous cortisone acetate on day 2 and day 6. From day 1 on, mice were treated daily with the antibiotic Baytril. On day 4, mice were anesthetized and intubated, and *A. fumigatus* (ATCC46645) forms were administered intratracheally (2 × 10^4^ in 50 μL 0.9% NaCl). For sphingosine inhalation, mice were immobilized in a restrainer and 1 mL of 500 µM sphingosine/0.5% OGP was given daily by inhalation via a PARI BOY SX inhalation device, starting on day 1. (**c**) Mice were treated as in (**b**), but were inhaled instead with placebo/0.5% OGP. (**d**) Lung sections from an untreated mouse were treated with DNase I to fragment the DNA and were used as positive controls. Shown is one representative lung section from *n* = 4 mice.

**Figure 4 ijms-23-15510-f004:**
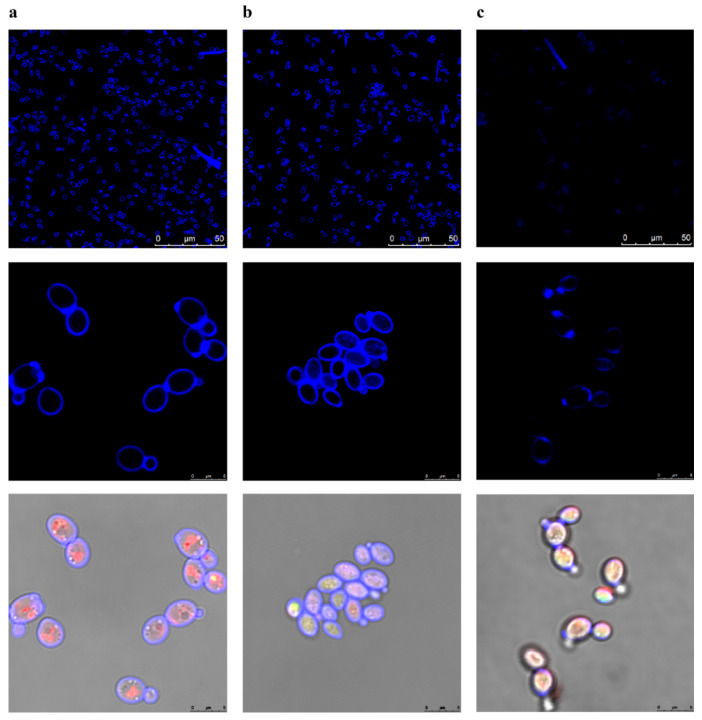
Cell wall integrity is not affected by treatment with sphingosine. *N. glabrataa* DSM70614 cells were treated for 4 h at 37 °C with sphingosine at a concentration of 2 × MIC, washed with phosphate-buffered saline (PBS) (pH 7.4), stained with 25 µM calcofluor white (upper row) or 10 µM FUN1 (middle row) for 30 min in the dark, and examined by fluorescence microscopy (lower row). (**a**) No treatment. (**b**) Sphingosine-treated (2 × MIC). (**c**) Treated with ethanol 70% *v*/*w* (positive control). Displayed are representative results of *n* = 3 independent experiments.

**Figure 5 ijms-23-15510-f005:**
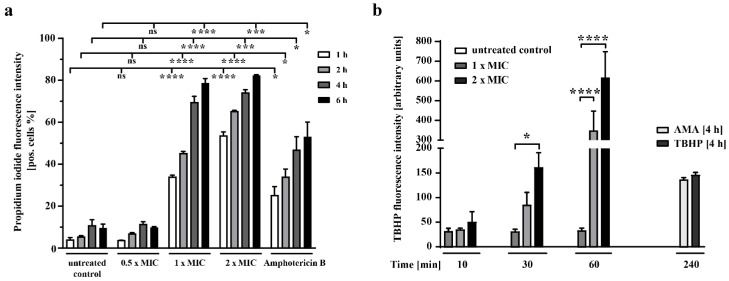
Treatment with sphingosine results in rapid formation of reactive oxygen species (ROS) and disruption of the cell membrane. (**a**) *N. glabrataa* DSM70614 cells were treated with 0.5 × MIC, 1 × MIC, or 2 × MIC sphingosine for 1, 2, 4, or 6 h as indicated. They were then washed with PBS and stained with 1 µg/mL propidium iodide. Fluorescence was measured with an Attune NXT flow cytometer. Shown are means ± standard deviation, *n* = 3. ns = not significant, * *p* < 0.05, *** *p* < 0.001, **** *p* < 0.0001, Dunnett’s multiple comparisons test. (**b**) Cells were exposed to various concentrations (1 × MIC, 2 × MIC) of sphingosine at 37 °C for 10, 30, or 60 min, or to 50 µM tertbutyl hydrogen peroxide (TBHP) and 10 µg/mL antimycin A (AMA) for 240 min as positive controls. Cellular production of ROS was detected 60 min after treatment with sphingosine. Shown are means ± standard deviation, *n* = 3. * *p* < 0.05; **** *p* < 0.0001, Dunnett’s multiple comparisons test.

**Figure 6 ijms-23-15510-f006:**
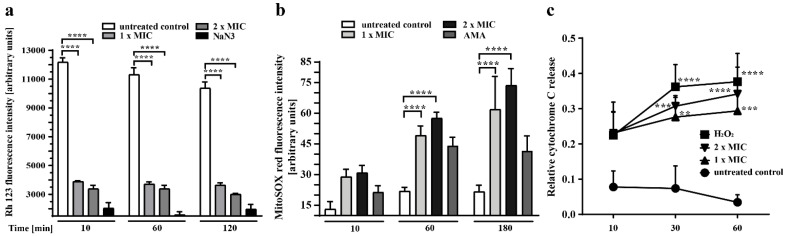
Treatment with sphingosine leads to the early depolarization of mitochondrial membrane potential (Δψm), to the generation of mitochondrial ROS, and directly to a release of cytochrome C. *N. glabrataa* DSM70614 cells were exposed to sphingosine (1 × MIC, 2 × MIC) and incubated for 10, 60, or 120 min. (**a**) Rh-123 was added at a final concentration of 25 µM for 10 min. Shown are means ± standard deviation, *n* = 3. **** *p* < 0.0001, Dunnett’s multiple comparisons test. (**b**) Cells treated with 10 µM antimycin A (AMA) were used as positive controls. All samples were stained with 5 µM MitoSOX Red for 15 min. Shown are means ± standard deviation, *n* = 3. **** *p* < 0.0001, Dunnett’s multiple comparisons test. (**c**) Mitochondria from *N. glabrataa* cells were freshly isolated, suspended in osmotic buffer, and incubated either with or without sphingosine (1 × MIC, 2 × MIC) for 10, 30, or 60 min at 25 °C. Mitochondria were centrifuged, and both the pellet and the supernatant were subjected to sodium dodecyl sulfate (15%) polyacrylamide gel electrophoresis (SDS-PAGE) and Western blotting with anti–cytochrome C antibodies. The intensity of signals from Western blot analysis was quantified with a Typhoon FLA 9000 biomolecular imager. Shown are means ± standard deviation, *n* = 3. ** *p* < 0.01, *** *p* < 0.001, **** *p* < 0.0001, Dunnett’s multiple comparisons test.

**Table 1 ijms-23-15510-t001:** Minimum inhibitory concentration (MIC) and minimum fungicidal concentration (MFC) of sphingosine against *Aspergillus* strain cells, according to European Committee on Antimicrobial Susceptibility Testing (EUCAST).

Fungal Strain	MIC µg/mL	MFC µg/mL
*A. fumigatus* ATCC204305	2	2
*A. fumigatus* ATCC46645	2	2
*A. fumigatus* 2446	2	2
*A. fumigatus* 2453	2	4
*A. fumigatus* 2150	2	2
*A. fumigatus* 2151	2	2
*A. fumigatus* 2040	2	>8
*A. niger* F19	4	4
*A. flavus* CM-1813	>8	-
*A. flavus* 2476	>8	-
*A. flavus* 2435	>8	-
*A. flavus* 2412	>8	-
*A. brasiliensis* 1988	>8	-
*A. tubingensis* 1884	>8	-
*A. tubingensis* 1885	>8	-

Two-fold dilutions of sphingosine were prepared in Roswell Park Memorial Institute (RPMI) 1640 + 3-(N-morpholino) propanesulfonic acid (MOPS) + 2% glucose (pH 7.0). The dilutions were mixed 1:1 with 100 µL of *Aspergillus* spp. cell suspensions in water (final working inoculum, 2 to 5 × 10^5^ cells/mL) in 96-well plates. Plates were incubated without agitation at 37 °C in ambient air for 48 h. The growth of *Aspergillus* strains was evaluated visually.

**Table 2 ijms-23-15510-t002:** Minimum inhibitory concentration (MIC) and minimum fungicidal concentration (MFC) against *yeast* strains according to European Committee on Antimicrobial Susceptibility Testing (EUCAST).

Yeast Strain	MIC µg/mL	MFC µg/mL
*N. glabrataa* DSM70614	1	1
*N. glabrataa* 195	2	4
*N. glabrataa* 196	2	2
*N. glabrataa* 160	2	4
*P. kudriavzevii* ATCC6258	4	4
*P. kudriavzevii* F31	8	>8
*P. kudriavzevii* 132	4	8
*P. kudriavzevii* 126	4	>8
*P. kudriavzevii* 201	4	4
*C. albicans* ATCC90028	>8	-
*C. albicans* 197	>8	-
*C. albicans* 204	>8	-
*C. albicans* 205	>8	-
*C. albicans* 207	>8	-
*C. albicans* 212	>8	-
*C. albicans* 223	>8	-
*C. tropicalis* DSM1346	>8	-
*C. parapsilosis* ATCC22019	>8	-

Two-fold dilutions of sphingosine were prepared in RPMI 1640 + MOPS + 2% glucose (pH 7.0) and was mixed 1:1 with 100 µL of *yeast* cell suspensions in water (final working inoculum, 2 to 5 × 10^5^ cells per mL) in 96-well plates. Plates were incubated without agitation at 37 °C in ambient air for 24. For quantification of *yeast* cells, absorbance was measured at 530 nm.

## Data Availability

Not applicable.

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
