# Peer review of "Sphingosine as a New Antifungal Agent against Candida and Aspergillus spp."

_ijms, 2022, doi:10.3390/ijms232415510_

Round 1

Reviewer 1 Report

This manuscript presents results from an interesting study of sphingosine as an antifungal agent. against various Aspergillus and Candida species. The introduction provides a nice summary of current areas of interest and prior findings. Overall, it is a nice manuscript. A few comments follow.

(Page numbering is not sequential...I used the page number from the file and the lines even though they restart.)

Page 4, lines 163-165 and Page 5, lines 197-199: These two statements allude to interesting findings related to weight loss and the appearance of lung tissue with various treatments. However, in both cases, it is noted that data are not given or are not shown. I suggest that data should be provided, possibly in the supplemental materials.

Page 3, lines 386-387: the manuscript states "...biofilm itself should not have been present..." and "...therefore not the cause...". Since biofilm presence was not specifically tested, this conclusion is a little too strong. It should not have been the cause but the experiments presented do not completely rule out the presence of biofilms.

Page 4, lines 464-465: the description of cardiolipin is not chemically accurate. Phosphatidate is the ion formed from phosphatidic acid and is thus not a molecule; glycerol is similarly not a molecule in the cardiolipin structure. One correct description of cardiolipin would be a "glycerol head group bound to two phosphatidyl moieties" or "two phosphatidyl moieties appended to a glycerol backbone".

Throughout: there are numerous minor spelling, format, or grammatical errors that should be addressed. For example, Page 6, line 209: "prophylactical" should be "prophylactic"; line 221: Student's t-test should be capitalized; etc.

Author Response

We thank for the very helpful comments. We have cusively presented the reviewer's text and inserted our answers in between. 

This manuscript presents results from an interesting study of sphingosine as an antifungal agent. against various Aspergillus and Candida species. The introduction provides a nice summary of current areas of interest and prior findings. Overall, it is a nice manuscript. A few comments follow.

(Page numbering is not sequential...I used the page number from the file and the lines even though they restart.)

Page 4, lines 163-165 and Page 5, lines 197-199: These two statements allude to interesting findings related to weight loss and the appearance of lung tissue with various treatments. However, in both cases, it is noted that data are not given or are not shown. I suggest that data should be provided, possibly in the supplemental materials.

Answer:

We added a new Fig. 1b with the weights of the mice. Unfortunately, we did not take macroscopic photographs of the lungs. Therefore, unfortunately, we cannot show this....but the findings were quite clear and, in our opinion, should be reported anyway. Therefore, we would like to leave this part as it is. Should the reviewer insist on it, we could of course remove the part, but in our opinion content would be lost...

Page 3, lines 386-387: the manuscript states "...biofilm itself should not have been present..." and "...therefore not the cause...". Since biofilm presence was not specifically tested, this conclusion is a little too strong. It should not have been the cause but the experiments presented do not completely rule out the presence of biofilms.

Answer:

We agree and have changed the sentences as follows: “However, our experiments were performed according to the EUCAST protocol and used planktonic cells; therefore, biofilm itself should not have been present at the time of sphingosine exposure and should not have caused the discrepancy. Nevertheless, we cannot exclude biofilm as a cause of the differential results because biofilm was not tested and the presence of biofilm could not be completely excluded.”

Page 4, lines 464-465: the description of cardiolipin is not chemically accurate. Phosphatidate is the ion formed from phosphatidic acid and is thus not a molecule; glycerol is similarly not a molecule in the cardiolipin structure. One correct description of cardiolipin would be a "glycerol head group bound to two phosphatidyl moieties" or "two phosphatidyl moieties appended to a glycerol backbone".

Answer:

We changed the description of cardiolipin accordingly.

Throughout: there are numerous minor spelling, format, or grammatical errors that should be addressed. For example, Page 6, line 209: "prophylactical" should be "prophylactic"; line 221: Student's t-test should be capitalized; etc.

Answer:

We had the entire manuscript revised for linguistic weaknesses/errors by a native speaker with 30 years’ medical editing experience: Flo Witte, PhD, of Bluegrass Editorial Services Team, Lexington, Kentucky, USA.

Reviewer 2 Report

Title: Sphingosine as a new antifungal agent for prophylactic treatment of pulmonary aspergillosis

Aspergillus species are emerging with increasing resistance to triazoles, thus studies evaluating new antifungal drugs have been more necessary.

This study presents an extensive methodology, however, there are several points in the manuscript that needs work.

1.       Revise the English.

2.       The manuscript shows data corresponding to some Candida species, but in the title, methodology, and discussion, these results no are explored.

3.       In introduction does not reference the importance about to investigated new antifungal agents, or resistance triazoles and other drugs.

4.       The authors present in the introduction aspects related to Candida genus, but it is confusing because the focus of the manuscript is Aspergillus.

5.       The nomenclature of the genus and species of the microorganisms should be according to international coding nomenclature for fungi and bacteria.

6.       In general, the introduction needs to better write.

7.       How were identified the Aspergillus and Candida isolates?

8.       We demonstrate that sphingosine is effective against a wide range of huma pathogenic fungi…. Wide range? What are these?

9.       Whilst C. glabrata is rather sensitive to sphingosine treatment, C. krusei, C. albicans, C. tropicalis and C. parapsilosis 380 strains were more resistant.. You could talk about resistance in this phrase? I believe that talk about "less sensitive" will be more adequate.

10.   …the ability to produce biofilm seems to correlate inversely to the sensitivity to sphingosine. These are not tested.

11.   In humans, an invasive aspergillosis, even in immunosuppressed, develops over several weeks to months. ii. due to the direct intratracheal application/nebulization the lung is affected in very short time; in the human situation invasive aspergillosis in the many most times does not affect the whole lung. Future studies are required to establish a curative model. Put the references.

12.   Discuss the results according to the sequence of results presented.

13.   Furthermore, it is possible that bronchial cells can rapidly metabolize sphingosine through increased expression of sphingosine kinases, therefore reducing its toxicity. Put the references.

14.   The authors do not discuss anything about Candida.

15.   Discuss prophylactic treatments of pulmonary aspergillosis

Author Response

We thank the reviewer for the very helpful comments.We have italicized the reviewer's text and inserted our comments directly:

  1.      Revise the English.

Answer:

We had the entire manuscript revised for linguistic weaknesses/errors by a native speaker with 30 years’ medical editing experience: Flo Witte, PhD, of Bluegrass Editorial Services Team, Lexington, Kentucky, USA.

  1.      The manuscript shows data corresponding to some Candida species, but in the title, methodology, and discussion, these results no are explored.

Answer:

We changed the title accordingly to “Sphingosine as a new antifungal agent against Candida and Aspergillus spp.”

  1.      In introduction does not reference the importance about to investigated new antifungal agents, or resistance triazoles and other drugs.

Answer:

We included the following statement in Introduction: “Guidelines suggest posaconazole as primary antifungal prophylaxis for patients at high risk of invasive aspergillosis [17]. Posaconazole has been shown to be superior to fluconazole; however, probable or proven breakthrough invasive fungal infections still occur in approximately 10% of patients treated with posaconazole [18]. Echinocandins may be used alternatively in case of azole resistance [19]. However, resistance of Aspergillus spp. to azoles and echinocandin can occur because of the use of these agents not only in clinical medicine but also in agriculture.”

Further, we changed the last paragraph in Introduction as following: “Because the therapeutic options for, e.g., invasive aspergillosis and also some candida infections are limited, and because of the development of resistance to azoles, echinocandin, and amphotericin B [24, 25], new therapeutic and prophylactic approaches are greatly needed. The present study was designed to determine whether sphingosine is suitable as a new antifungal treatment. We tested and characterized the efficacy of sphingosine in vitro under culture conditions and in a murine model of pulmonary aspergillosis.

  1.      The authors present in the introduction aspects related to Candida genus, but it is confusing because the focus of the manuscript is Aspergillus.

Answer:

Because we have changed the title as suggested and thus also brought Candida spp. into focus, it now seems more understandable to us why the introduction also covers aspects of Candida. 

  1.      The nomenclature of the genus and species of the microorganisms should be according to international coding nomenclature for fungi and bacteria.

Answer:

We changed the designations according to the international coding nomenclature for fungi and bacteria.

  1.      In general, the introduction needs to better write.

Answer:

We had the entire manuscript revised for linguistic weaknesses/errors by a native speaker (see above). Furthermore, we included in the introduction additions regarding your point 3.

  1.      How were identified the Aspergillus and Candida isolates?

Answer:

We included the following statement in Methods: “Yeast isolates were identified by matrix-assisted laser desorption/ionization time-of-flight (MALDI-TOF) mass spectrometry (VITEK; bioMérieux, Nürtlingen, Germany). A. fumigatus was identified by the typical macro-and micromorphology and growth at 50 °C. Bacteria other than A. fumigatus spp. were identified on the basis of their morphology and on analysis of the internal transcribed spacer (ITS) 1 and 2 DNA sequences.

  1.      We demonstrate that sphingosine is effective against a wide range of huma pathogenic fungi…. Wide range? What are these

Answer:

We have replaced the words "wide range" with "some of the strains tested (Tables 1 and 2)"

  1.      Whilst C. glabrata is rather sensitive to sphingosine treatment, C. krusei, C. albicans, C. tropicalis and C. parapsilosis 380 strains were more resistant.. You could talk about resistance in this phrase? I believe that talk about "less sensitive" will be more adequate.

Answer:

We agree…we replaced “resistant” with “less sensitive”. In the paragraph we have also made the consequential analogous changes.

  1.  …the ability to produce biofilm seems to correlate inversely to the sensitivity to sphingosine. These are not tested.

Answer:

We agree and changed the wording as follows: “However, our experiments were performed according to the EUCAST protocol and used planktonic cells; therefore, biofilm itself should not have been present at the time of sphingosine exposure and should not have caused the discrepancy. Nevertheless, we cannot exclude biofilm as a cause of the differential results because biofilm was not tested and the presence of biofilm could not be completely excluded.” 

  1.  In humans, an invasive aspergillosis, even in immunosuppressed, develops over several weeks to months. ii. due to the direct intratracheal application/nebulization the lung is affected in very short time; in the human situation invasive aspergillosis in the many most times does not affect the whole lung. Future studies are required to establish a curative model. Put the references.

Answer: We added following reference: Doffman, S.R.; Agrawal, S.G.; Brown, J.S. Invasive Pulmonary Aspergillosis. Expert. Rev. Anti-Infect. Ther. 2005, 3, 613-627; DOI:10.1586/14787210.3.4.613.

  1.  Discuss the results according to the sequence of results presented.

Answer:

This point relates to the paragraph that discusses the mode of action of sphingosine. We have rephrased the respective paragraph accordingly: 

“Regarding our studies aimed at elucidating the mode of action of sphingosine, we selected the reference strain of N. glabrataaDSM70614, which is sensitive to sphingosine treatment (Table 2) as experimental system to elucidate the mode of action of sphingosine. We found that treatment with sphingosine does not affect cell wall integrity. We found that the cell membrane is disrupted by staining with PI, and this disruption increased substantially during incubation within 6 hours. In addition, we found a marked increase in the amount of cellular ROS within one hour after treatment. We also found that treatment with sphingosine leads to early depolarization of the MtMP (Δψm) and to mitochondrial ROS generation within minutes. We also found a direct effect of sphingosine on mitochondria leading to the release of cytochrome C as a hallmark of apoptosis. In summary, our findings suggest that the mode of action of sphingosine on fungi is due to a direct effect on mitochondria and that the disruption of the cell membrane may be a later, secondary event in the frame of apoptosis.”

  1.  Furthermore, it is possible that bronchial cells can rapidly metabolize sphingosine through increased expression of sphingosine kinases, therefore reducing its toxicity. Put the references.

This has not been tested so far. Therefore we changed the wording as follows: “It is also conceivable that bronchial epithelium metabolizes sphingosine more rapidly than other tissues do because of increased expression of, e.g., sphingosine kinases, which reduce the toxicity of sphingosine. However, this hypothesis has not been investigated so far and remains speculative.”

  1.  The authors do not discuss anything about Candida.

Answer:

We included the following paragraph in the discussion: “Yeasts such as C. albicans, N. glabrataa, P. kudriavzevii, C. tropicalis, and C. parapsilosis are the most common cause of vaginal or mucosal infections and may, among immunosuppressed patients, enter the bloodstream and cause deep tissue or systemic infections [20, 21]. Systemically applied azoles, echinocandin, and locally applied polyenes are used as prophylaxis against various diseases. Our in vitro findings show that N. glabrataa and P. kudriavzevii, both known to have an intrinsic resistance to azoles (61), are sensitive to sphingosine treatment. It is quite conceivable that prophylactic or therapeutic treatment by local application of sphingosine in creams, for example, could have additional treatment applications. However, the present project did not pursue this option, and the possibility remains untested.”

  1.  Discuss prophylactic treatments of pulmonary aspergillosis

Answer:

Please see the answer for your point 3. Here, we have now gone into more detail about the prophylactic treatment of pulmonary aspergillosis.

We further added to the discussion following paragraph: “Our data suggest that the therapeutic targets of sphingosine are particularly the mitochondria of fungi. This distinguishes sphingosine from other drugs used in routine clinical practice, such as azoles, polyenes and echinocandins. Thus, it can be speculated that a prophylactic treatment with inhaled sphingosine will not lead to a development of resistance against the drugs already used in clinical routine. This could therefore be an important advantage for the use of sphingosine as a prophylactic agent, since in case of a breakthrough infection the whole range of proven substances would still be available. However, this was not investigated in the present study and thus remains speculative.”